# Therapeutic Endoscopic Ultrasound for Complications of Pancreatic Cancer

**DOI:** 10.3390/cancers16010029

**Published:** 2023-12-20

**Authors:** Samuel Han, Georgios I. Papachristou

**Affiliations:** Division of Gastroenterology, Hepatology, and Nutrition, The Ohio State University Wexner Medical Center, Columbus, OH 43210, USA; samuel.han@osumc.edu

**Keywords:** pancreatic cancer, choledochoduodenostomy, endoscopic ultrasound, hepaticogastrostomy, biliary obstruction, gastric outlet obstruction

## Abstract

**Simple Summary:**

Complications of pancreatic cancer include the development of blockage of the bile duct preventing drainage of the liver, blockage of the stomach and duodenum preventing the passage of food and increase in pain levels. Endoscopic ultrasound is a procedure performed through the mouth where a flexible tube with a camera and an ultrasound probe attached allow for visualization of organs around the stomach and small intestine, such as the pancreas, the bile duct, and the liver. It was originally designed to help diagnose pancreatic cancer, but with advances in technology, it can now be used to treat the aforementioned complications without performing surgery. This paper reviews the new advances in technologies and techniques performed through endoscopic ultrasound with an emphasis on clinical trials that have been done to provide an overview of the safety and effectiveness of these procedures.

**Abstract:**

Progression of pancreatic adenocarcinoma can result in disease complications such as biliary obstruction and gastric outlet obstruction. The recent advances in endoscopic ultrasound (EUS) have transformed EUS from a purely diagnostic technology to a therapeutic modality, particularly with the development of lumen-apposing metal stents. In terms of biliary drainage, EUS-guided choledochoduodenostomy and EUS-guided hepaticogastrostomy offer safe and effective techniques when conventional transpapillary stent placement via ERCP fails or is not possible. If these modalities are not feasible, EUS-guided gallbladder drainage offers yet another salvage technique when the cystic duct is non-involved by the cancer. Lastly, EUS-guided gastroenterostomy allows for an effective bypass treatment for cases of gastric outlet obstruction that enables patients to resume eating within several days. Future randomized studies comparing these techniques to current standard-of-care options are warranted to firmly establish therapeutic EUS procedures within the treatment algorithm for this challenging disease.

## 1. Introduction

Pancreatic ductal adenocarcinoma (PDAC) carries a dismal 5-year survival rate and represents the third leading cause of cancer-related mortality in the United States and the seventh leading cause worldwide [1]. Depending on the location of the primary tumor and spread of the disease, complications of PDAC include the development of biliary obstruction, gastric outlet obstruction, and pain, among others. Traditionally, endoscopic ultrasound (EUS) has played a significant role in the diagnosis and local staging of PDAC [2]. With the development of fine-needle biopsy (FNB) in particular, diagnostic yields of EUS have approached 99% in addition to offering the ability to sample lymph nodes, metastases to the liver, and ascitic fluid [3,4].

The last decade has seen the evolution of EUS from a primarily diagnostic modality to a therapeutic tool for the management of PDAC. Technological advances in devices such as lumen-apposing metal stents (LAMS) in particular, have enabled endoscopists to offer treatment methods for complications of PDAC as an alternative to surgical or interventional radiology-based options.

This review will focus on the latest advances in therapeutic EUS for the treatment of PDAC-related complications (Table 1). In presenting data regarding the safety and efficacy of these modalities, we hope to increase awareness of the growing use of EUS as an adjunctive tool for both medical and surgical oncologists in the management of this deadly disease.

## 2. EUS-Based Treatment Modalities

### 2.1. EUS-Guided Biliary Drainage

Jaundice resulting from biliary obstruction remains the most common presentation of PDAC involving the head of the pancreas [5]. Currently, biliary decompression via endoscopic retrograde cholangiopancreatography (ERCP) with a biliary stent, typically a cylindrical self-expanding metal stent (SEMS), represents the first-line treatment option for biliary obstruction owing to its high success rates and low adverse event rates [6]. Additionally, ERCP with stent placement can be performed at the same time as the diagnostic EUS with FNB, potentially saving the patient from additional procedures. Despite this, ERCP may not be feasible in up to 14% of cases such as when endoscopic access to the major papilla is blocked by extension of the cancer (duodenal obstruction) or when the cancer invades the ampulla, causing obliteration of the bile duct orifice [7,8,9,10]. In these circumstances, percutaneous biliary drainage was traditionally offered, but EUS-guided biliary drainage now presents a minimally invasive alternative treatment in select cases.

### 2.2. EUS-Guided Choledochoduodenostomy

EUS-guided Choledochoduodenostomy (EUS-CD) entails the placement of a stent from the duodenum (typically the duodenal bulb) into the common bile duct. Advantages of EUS-CD over ERCP include eliminating the risk of post-procedure pancreatitis given the lack of contact with the major papilla and the pancreatic duct, as well as a reduction in the risk of stent occlusion from tumor growth as the stent is typically placed above the level of the obstruction, as opposed to through the obstruction when placed via ERCP.

Initial studies on EUS-CD examined the use of transmural cylindrical SEMS for biliary drainage. In this technique, the bile duct is punctured under EUS-guidance with a needle (usually 19 G in size) and a cholangiogram is performed. A long (>450 cm length) guidewire is then advanced towards the hilum and the needle is withdrawn. The choledochoduodenostomy tract is then dilated (with either a balloon, passage dilator, or using cautery) and the SEMS is subsequently advanced towards the hilum with the distal end of the stent deployed within the duodenum. In regard to tract dilation, the use of cautery appears to pose a higher risk of bleeding than using a balloon or passage dilator [11,12,13]. Several randomized trials have compared EUS-CD using this technique with ERCP in patients with malignant distal biliary obstruction. Two multicenter randomized trials from Korea involving a combined 155 patients (70.0% PDAC) both found no difference between EUS-CDS and ERCP in technical and clinical success [9,14]. EUS-CD, however, was found to have a lower rate of adverse events (namely no events of post-procedure pancreatitis) and a lower rate of reintervention compared to ERCP. A single-center randomized trial from the US involving 67 patients (95.5% PDAC—22.4% with metastases) similarly found no difference in technical success and clinical success between EUS-CD and ERCP [10]. In contrast to the two aforementioned studies, however, there was no difference in adverse events between the two treatments.

The development of the LAMS and specifically the electrocautery-enhanced LAMS (Hot AXIOS, Boston Scientific, Marlborough, MA, USA; Hot Spaxus, Taewoong Medical, Gimpo, Republic of Korea; Hanarostent Hot Plumber Z-EUS IT, M.I. Tech, Seoul, Republic of Korea) has greatly simplified the process of performing an EUS-CD (Figure 1) [15]. The electrocautery-enhanced LAMS enables the endoscopist to puncture the bile duct and deploy the stent in a single step, not only saving time, but also reducing the risk of adverse events such as bile leakage and infection that can occur when performing the multiple steps required with the traditional EUS-CD. A recent randomized trial involving multiple sites from different continents compared EUS-CD utilizing a LAMS with ERCP for biliary decompression in patients with unresectable malignant distal biliary obstruction (*n* = 155, 96.1% PDAC) [7]. EUS-CD had a superior technical success rate (96.2% vs. 76.3%) with no difference in 1-year stent patency rates or other adverse events. The mean survival was 232.2 ± 190.9 days with EUS-CD and 202.6 ± 176.3 days with ERCP. As expected, EUS-CD had a significantly shorter procedural time (10 vs. 25 min) than ERCP, highlighting the primary advantages of EUS-CD over ERCP; the ability to perform biliary drainage when the major papilla is inaccessible and quicker procedure times. Another recent multicenter randomized trial from Canada (*n* = 144) compared EUS-CD with LAMS with ERCP in patients with periampullary cancer (90.3% PDAC—35.2% borderline resectable/locally advanced, 64.8% unresectable) [8]. While there was no difference in technical success rates (90.4% EUS-CD vs. 83.1% ERCP), EUS-CD had a shorter procedure time (14 vs. 23 min) with no difference in adverse events or stent dysfunction. In addition to these trials, several other studies have corroborated the high success rates of using LAMS for EUS-CD. A large retrospective multicenter study (*n* = 256, 75% PDAC) from Italy examined the use of EUS-CD with LAMS after a failed ERCP, finding a technical success and clinical success rate of 93.3% and 96.2%, respectively [16]. Food occlusion resulting in stent obstruction occurred in 6.7% of patients and bleeding occurred in 1.7% of patients. Another retrospective multicenter study (*n* = 52, 82.7% PDAC) from France found a technical success rate of 88.5% with a clinical success rate of 100% with a mean procedure duration of 10 min [17]. Long-term adverse events primarily consisted of recurrence of jaundice arising from tumoral obstruction (7.7%) and food occlusion (3.9%). Notably, a recent single-center prospective study (*n* = 22, 91% PDAC—46% resectable, 27% locally advanced, 27% metastatic) from the Netherlands found that despite high technical and clinical success rates using a LAMS for EUS-CD as the primary biliary drainage strategy, a high proportion of subjects (55%) experienced some degree of stent dysfunction including compression of the LAMS within the bile duct (20%) and food impaction (10%) [18]. The majority of these cases of stent dysfunction were treated successfully with repeat endoscopic intervention.

### 2.3. EUS-Guided Hepaticogastrostomy

In cases where duodenal invasion of the cancer precludes access to the major papilla and the duodenal bulb for performing an EUS-CD, EUS-guided hepaticogastrostomy (EUS-HG) offers another technique for draining the biliary system. From the stomach, an intrahepatic duct (typically left hepatic duct segment 2 or 3; B2 or B3) is punctured with a needle (usually 19 G). A cholangiogram is performed and a long (>450 cm) angled guidewire advanced towards the common bile duct. After the removal of the needle, the hepaticogastrostomy tract is typically dilated before a stent is advanced into the intrahepatic duct with release of the proximal end of the stent safely within the stomach. As with EUS-CD, dilation of the hepaticogastrostomy tract can be performed using a balloon or passage dilator or cautery with cautery-enhanced dilation being associated with higher bleeding rates [11,19]. Newer stent designs can bypass the need for dilation, however, utilizing a tapered tip on the stent introducer to advance the delivery system directly into the hepatic duct. Similar to EUS-CD, EUS-HG eliminates the risk of post-procedural pancreatitis but also can be offered for biliary drainage in patients with surgically altered anatomy where the duodenum is not accessible endoscopically. Adverse events specific to the EUS-HG include the risk of stent migration into the peritoneum (due to the free space between the stomach and the liver and constant motion of the liver during respiration) and mediastinitis (due to the risk of puncturing the left hepatic duct from the esophagus) [20]. In patients with PDAC where the obstruction is typically within the distal bile duct, antegrade stenting of the obstruction can also be performed in conjunction with the EUS-HG. If performing antegrade stenting, the guidewire is manipulated to traverse the distal stricture into the duodenum and a SEMS is deployed in antegrade fashion across the stricture into the duodenum. A second stent is then placed through the hepaticogastrostomy tract in standard fashion to connect the left hepatic duct with the stomach, thus providing two access points for biliary drainage.

In a two-center retrospective study from Japan (*n* = 110, 50% PDAC—35% metastatic), EUS-HG carried high technical and clinical success rates of 100% and 94%, respectively [21]. Recurrent biliary obstruction occurred in 33% of patients at a median of 6.3 months and the majority were successfully treated with endoscopic reintervention. Most adverse events were mild in severity with peritonitis developing in 4.4% of cases. Another multicenter retrospective study (*n* = 152, 40.8% PDAC) from Japan compared the use of SEMS with plastic stents for EUS-HG, finding that SEMS had a longer stent patency duration (646 days vs. 202 days) with no difference in clinical success rates [22]. A multicenter prospective study (*n* = 49, 22.4% PDAC) found that EUS-HG combined with antegrade biliary stenting carried a technical success rate of 85.7% [23]. Stent obstruction (14.9%) accounted for the most frequent adverse event and the mean stent patency duration was 320 days. In a single-center retrospective study (*n* = 57, 100% PDAC), EUS-HG performed with antegrade biliary stenting had a technical and clinical success rate of 91.2% with a median procedural time of 25 min [24]. Recurrent biliary obstruction occurred in 30.8% of patients at a median of 245 days and endoscopic reintervention was successful in 100% of cases.

Several studies have compared EUS-CD with EUS-HG for biliary drainage. A retrospective international multicenter study (*n* = 121, 54% PDAC) found that while the technical and clinical success rates were equivalent between the two techniques, EUS-CD was associated with a shorter hospitalization stay [25]. There was no difference in the adverse event rate between the two modalities as well. A single-center retrospective study (*n* = 39, 82.1% PDAC) from Japan in patients with both biliary and duodenal obstruction (treated via duodenal stenting) found that EUS-HG was associated with a longer biliary stent patency (133 days vs. 43 days) and duodenal stent patency (113 days vs. 34 days) than EUS-CD [26].

### 2.4. EUS-Guided Gallbladder Drainage

In cases where the cancer does not involve the cystic duct, EUS-guided gallbladder drainage (EUS-GB) offers a salvage modality to relieve biliary obstruction when ERCP fails. This technique entails the placement of a stent from either the gastric antrum or the duodenal bulb into the gallbladder. LAMS have simplified this into a single-step procedure which has proven effective as a non-surgical management of cholecystitis [27]. As a rescue biliary drainage modality, a multicenter study (*n* = 48, 85.4% PDAC) from Italy found an 81.3% clinical success rate with a 10.4% adverse event rate (three cases of stent dysfunction consisting of stent dislodgement, migration, and occlusion) [28]. In a multicenter retrospective study (*n* = 28) from the US, EUS-GB had a clinical success rate of 93% with the most common adverse event being food impaction of the stent (10.7%) [29]. A retrospective study from Japan (*n* = 12, 50% PDAC) similarly found a clinical success rate of 91.7% with one case of stent dysfunction arising from occlusion of the cystic duct from tumor progression [30].

A specific concern after transpapillary SEMS placement during ERCP is the risk of cholecystitis which can occur in up to 16% of cases, particularly if the tumor involves the cystic duct orifice [31,32]. A randomized trial (*n* = 44, 52.3% PDAC) from Ecuador compared prophylactic EUS-GB after SEMS placement during ERCP with SEMS placement alone, finding a significantly lower rate of cholecystitis in the EUS-GB arm (0% vs. 22.7%) [33]. A retrospective study (*n* = 30, 33.3% PDAC) also demonstrated that EUS-GB offers a viable treatment strategy for cholecystitis occurring after SEMS placement with a 96% clinical success rate [31].

### 2.5. Comparison of EUS-Guided Biliary Drainage with Percutaneous Biliary Drainage

A multicenter randomized trial (*n* = 66, 36.4% PDAC) from Korea compared EUS-guided biliary drainage (EUS-CD or EUS-HG) with percutaneous transhepatic biliary drainage (PTBD) in patients where ERCP had failed [34]. There was no difference in clinical success between the two modalities, but PTBD carried a higher adverse rate (31.2% vs. 8.8%) consisting primarily of cholangitis. Furthermore, PTBD required a higher number of re-interventions and was associated with a longer hospital stay. A retrospective study (*n* = 95, 85% PDAC) from France comparing EUS-CD using LAMS with PTBD found EUS-CD to have a significantly higher clinical success rate and overall survival rate in addition to a lower adverse event rate, shorter hospital stay, and lower overall cost [35]. Similarly, a retrospective multicenter experience (*n* = 86, 54.7% PDAC) from the US found a higher clinical success rate with EUS-CD compared to PTBD in addition to a lower rate of reintervention (10.7% vs. 77.6%) [36]. Lastly, a network meta-analysis (*n* = 217, five randomized trials) found that PTBD was inferior to both EUS-CD and EUS-HG in terms of clinical success, although no significant difference was found in adverse events [37].

While there is currently no evidence-based algorithm for biliary drainage, at our center, we follow a multidisciplinary approach to patients with PDAC-related biliary obstruction that incorporates patient anatomy and cancer stage (Figure 2).

### 2.6. EUS-Guided Gastroenterostomy

Gastric outlet obstruction (GOO) can occur in up to 20% of patients with PDAC, compromising nutritional status and quality of life in these patients [38]. Treatment has typically entailed the placement of a duodenal SEMS or surgical gastroenterostomy. Analogous to EUS-CD, the development of the EUS-guided gastroenterostomy (EUS-GE) has been greatly simplified with the development of LAMS. The jejunum distal to the site of obstruction is typically irrigated with a mixture of saline, contrast, and methylene blue and under EUS visualization, a loop of jejunum is punctured with the LAMS catheter with deployment of the distal flange of the stent in the jejunum and proximal flange of the stent in the gastric lumen (Figure 3). In this manner, EUS-GE bypasses the site of the obstruction completely, eliminating the risk of tumor invasion into the stent as seen with duodenal stents.

A prospective multicenter study (*n* = 64, 35.9% PDAC, all unresectable) from Spain found a 98.5% technical success rate in performing EUS-GE with 75.4% of patients able to eat 7 days post-procedure [39]. Furthermore, at 30 days post-procedure, 84.4% of patients tolerated a diet with investigators finding a significant improvement in quality of life at this time point. A prospective cohort study (*n* = 70, 75.7% PDAC, 60% metastatic) from Italy found a clinical success rate of 97.1% with recurrence of symptoms in 7.6% of patients [40]. Patients were able to resume eating solid food at a median of 1.5 days and resume chemotherapy at a median of 19 days. The most common adverse event was bleeding (5.7%) with the severe adverse events entailing two cases of stent misdeployment that required surgical retrieval.

A recent international multicenter randomized trial (*n* = 97, 45.4% PDAC) compared EUS-GE with duodenal stenting in patients with unresectable disease [41]. Treatment with EUS-GE led to a significantly lower 6-month reintervention rate (4% vs. 29% with both reinterventions in the EUS-GE arm occurring in subjects who had actually received a duodenal stent in an intention-to-treat analysis) compared to duodenal stenting. EUS-GE was also associated with longer stent patency (HR: 0.13, 95% CI 0.08–0.22) and improved eating (mean of 2.41 vs. 1.90 on gastric outlet obstruction score). There were also no differences in adverse events between the two treatments and specifically no cases of stent misdeployment in the EUS-GE arm. A large multicenter retrospective study (*n* = 176, 53.4% PDAC) from Spain also compared EUS-GE with duodenal stenting, finding no difference in the technical or clinical success rates and adverse events [42]. EUS-GE, however, had an improved stent patency rate at 3 months (92.2% vs. 80.6%) compared to duodenal stenting. A meta-analysis comparing EUS-GE and duodenal stenting included five retrospective studies involving 659 patients also found no difference in clinical or success rates [43]. EUS-GE, however, had a significantly lower rate of reintervention (4% vs. 23.6%) with no difference in adverse events.

No randomized trials thus far have compared EUS-GE with surgical gastroenterostomy. In an international multicenter propensity score comparison utilizing retrospective data (*n* = 125, 40.8% PDAC), EUS-GE had a shorter median time to oral intake (1 day vs. 3 days) and shorter hospitalization (4 days vs. 8 days) compared to laparoscopic gastroenterostomy [44]. There were no differences in technical or clinical success rates, but EUS-GE had a significantly lower adverse event rate (2.7% vs. 27%). A meta-analysis including seven studies with 625 patients found EUS-GE to be superior to surgical gastroenterostomy in terms of clinical success (OR: 4.73, 95% CI 1.83–12.25), post-procedure length of stay, and adverse event rate (OR: 0.2, 95% CI: 0.1–0.37) [45].

Relatedly, several studies have examined the concept of a double bypass where EUS-guided biliary drainage is performed with EUS-GE during the same procedure. A retrospective multicenter study (*n* = 154, 63.6% PDAC) from the Netherlands compared the combination of EUS-guided biliary drainage and EUS-GE with performing a surgical gastroenterostomy and hepaticojejunostomy [46]. Although patients treated via endoscopic double bypass had a higher comorbidity index, endoscopic treatment had a higher clinical success rate (94.1% vs. 82.2%) and shorter time to oral intake (0 vs. 6 days) than patients in the surgical treatment arm. Furthermore, EUS-biliary drainage combined with EUS-GE had a lower adverse event rate (11.3% vs. 34.7%) and shorter hospitalization (4 days vs. 13 days). Another multicenter study (*n* = 93, 73% PDAC—57% metastatic) from Europe compared different combinations of endoscopic treatments with permutations including EUS-CD, EUS-HG, transpapillary SEMS via ERCP, duodenal stenting, and EUS-GE [47]. The combination of EUS-GE with EUS-CD carried a clinical success rate of 88.9% while the EUS-GE with EUS-HG combination had an 83.3% clinical success rate. The lowest clinical success rate (60%) was seen in the combination of EUS-CD with duodenal stenting. A two-center retrospective study (*n* = 23, 69.6% PDAC) examined the combination of EUS-GE with EUS-HG, finding that all patients tolerated a soft diet post-procedure and 72.7% of patients had an adequate decrease in bilirubin [48]. Reinterventions were required in three (14%) patients to address recurrence jaundice.

### 2.7. Afferent Loop Syndrome

Afferent loop (limb) syndrome occurs when the afferent limb (post-Whipple) becomes obstructed, resulting in the signs and symptoms of biliary obstruction. While traditionally managed surgically via repeat anastomosis or bypass, EUS-guided enteroenterostomy (EUS-EE) offers an alternative endoscopic treatment option. Drawing upon the same principles of an EUS-GE, EUS-EE entails placing a LAMS connecting the afferent limb with either the stomach or the jejunum. A multicenter retrospective study (*n* = 45, 68.9% PDAC) observed a technical success rate of 95.6% and clinical success rate of 91.1% with EUS-EE [49]. Recurrent cholangitis occurred in 14.6% of cases, primarily due to stent occlusion (7.3%) and cancer recurrence at the hepaticojejunal anastomosis (4.9%) with endoscopic reintervention successfully performed in all cases. An earlier multicenter retrospective study (*n* = 18, 77.8% PDAC) found a clinical success rate of 88.9% with reinterventions (all endoscopic) required in 16.7% of cases [50]. Lastly, EUS-HG has also been examined as a treatment option for afferent limb syndrome by allowing for biliary drainage through the stomach. In a two-center retrospective study (*n* = 17, 35.3% PDAC), EUS-HG had a 100% clinical success rate with greater decreases in bilirubin seen compared to PTBD [51]. Furthermore, no interventions were required in the EUS-HG arm.

### 2.8. EUS-Guided Celiac Plexus Neurolysis

Pain control remains a significant challenge in the management of patients with PDAC, for which EUS-guided celiac plexus neurolysis (EUS-CPN) offers a potential therapy. This technique entails the injection of alcohol through a needle under EUS guidance to essentially perform the chemical destruction of the celiac ganglia and is primarily used in patients with locally advanced or metastatic disease. Several randomized trials (*n* = 398, 100% PDAC) have examined EUS-CPN with the majority finding improvement in pain levels at 1–3 months follow-up [52,53,54,55,56,57]. Adverse events are typically mild and include short-term diarrhea (4–15%) and postprocedural hypotension (1%) [58]. There exist a variety of modifications including identifying the celiac ganglia and directly injecting into the ganglia (celiac ganglia neurolysis), which was found in a randomized study (*n* = 110, 100% PDAC—83.4% T4, 52.7% M1) to be associated with decreased survival (26% vs. 42% 1-year survival) with no comparative benefit in pain compared to standard CPN [53]. Additionally, alcohol can be injected at the celiac trunk or both sides of the celiac trunk (bilateral injection), but a single-center randomized study (*n* = 50, 100% PDAC) found no difference in pain relief between the two approaches [57]. Alternatively, a small, randomized trial (*n* = 26, 100% PDAC—46.2% Stage IV) compared EUS-CPN with EUS-guided radiofrequency ablation (EUS-RFA) of the celiac plexus, finding that EUS-RFA led to greater pain relief than traditional CPN [56]. Further studies are needed, however, to validate the safety and efficacy of EUS-RFA for this indication, particularly given that the specific RFA device is no longer commercially available. Importantly, a randomized trial comparing EUS-CPN with a placebo is warranted to understand the true efficacy of this treatment.

### 2.9. EUS-Guided Radiofrequency Ablation

EUS-RFA offers a local treatment modality for advanced PDAC whereby alternating current at a frequency of 350–500 kHz is applied locally to the tumor via needle catheter. This modality is thought to produce coagulative necrosis of the mass while also directly burning the lesion with an in vivo study demonstrating reduced tumor progression and remodeling of the tumor microenvironment after RFA [59,60]. By inducing necrosis, cellular debris are released which are thought to stimulate an immune response including elevated levels of IL-6, HGF, and VEGF that may help target PDAC [59,61]. Small series have reported the safety and efficacy of this technique with one series of eight patients (87.5% PDAC—100% locally advanced, mean tumor size of 3.6 cm) from Italy reporting a 100% technical success rate with no adverse events and a 30% reduction in tumor size at 1 month follow-up [62]. Similarly, another series from Italy (*n* = 10, 100% unresectable, non-metastatic PDAC) found a 100% technical success rate with no major adverse events [63]. Most recently, a prospective study from the U.S. (*n* = 10, 70% locally advanced, 30% metastatic) found no adverse events and reported that two patients were able to subsequently undergo R0 pancreaticoduodenoectomy [59]. Of note, EUS-RFA has been effectively utilized in other pancreatic lesions such as insulinomas with high success rates [64,65].

## 3. Impact of Therapeutic EUS on Subsequent Surgery

While many of the studies presented included patients with unresectable PDAC, one of the critical questions remaining is how transmural stenting via therapeutic EUS affects the candidacy of potential surgical candidates. In other words, can therapeutic EUS serve safely and effectively as a bridge to surgery? In examining the limited data on this topic, one randomized trial comparing EUS-CD with ERCP for biliary drainage found that EUS-CD did not impede the subsequent Whipple procedure in five patients (100% PDAC) [10]. A randomized trial comparing EUS-CD using LAMS with ERCP in Stage III/IV cancer found no difference in the rates of subsequent Whipple procedures (8.2% EUS-CD vs. 5.6% ERCP) and no difference in surgical outcomes [8]. An international multicenter retrospective study (*n* = 145, 68.3% PDAC) found that higher rates of successful subsequent surgery (97% vs. 83%) were seen in EUS-guided biliary drainage (*n* = 58) via EUS-CD and EUS-HG compared to ERCP [66]. Additionally, patients who received EUS-guided biliary drainage had lower rates of reintervention before surgery (9% vs. 38%) compared to ERCP. Similarly, a retrospective multicenter study (*n* = 156, 53.8% PDAC) from Europe compared biliary drainage via EUS-CD using a LAMS with SEMS placement via ERCP [67]. All patients successfully went on to receive the Whipple procedure. Patients treated via EUS-CD, however, had a lower rate of overall (endoscopic and surgical) adverse events and surgical adverse events and a shorter length of stay. Furthermore, there was no difference in the R0 rate or overall survival. In regard to GOO, limited case reports have shown that pancreaticoduodenectomy is still feasible after EUS-GE in patients with PDAC [48,68].

## 4. Conclusions and Future Directions

In summary, therapeutic EUS has opened the doors for minimally invasive treatment options for patients with biliary obstruction or gastric outlet obstruction from PDAC. The development of the LAMS in particular has greatly facilitated performing EUS-CD and EUS-GE, thus enabling one-step stent placement, which can reduce procedure time and decrease the rate of adverse events. While the majority of studies on LAMS have utilized one particular device (Hot AXIOS, Boston Scientific, Marlborough, MA, USA), it should be noted that other LAMS devices have also been examined (Hot Spaxus, Taewoong Medical, Kimpo, Republic of Korea) for these indications [69,70]. While data reporting regarding these procedures and their adverse events remains an ongoing process, one must remember that all the reported studies involve expert centers with highly experienced endoscopists and for the time being, we recommend referral to tertiary centers with a high comfort level for these procedures.

Future studies and continued innovation are needed in this arena to best understand where these therapeutic options stand within the treatment algorithm of pancreatic cancer. In particular, continued development in stent technology is needed to further simplify these procedures (especially when performing EUS-HG), reduce the risk of stent migration and occlusion, and lengthen in-dwelling times. Lastly, randomized trials comparing EUS-GE with surgical gastroenterostomy are needed as are data regarding surgical feasibility after these therapeutic EUS procedures.

## Figures and Tables

**Figure 1 cancers-16-00029-f001:**
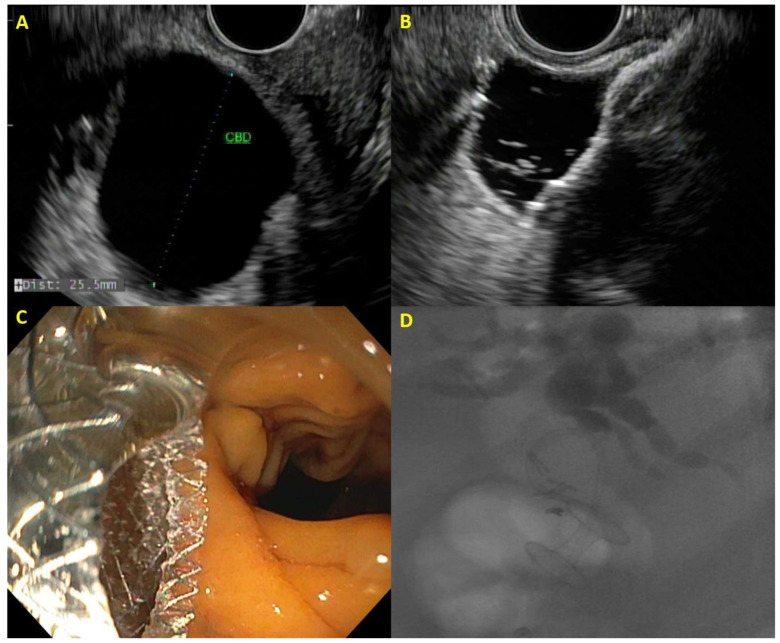
Example of EUS-guided choledochoduodenostomy with (**A**) dilated bile duct on EUS, (**B**) distal flange of lumen-apposing metal stent deployed within the bile duct, (**C**) proximal flange of lumen-apposing metal stent draining bile in the duodenum, and (**D**) cholangiogram demonstrating correct placement of choledochoduodenostomy.

**Figure 2 cancers-16-00029-f002:**
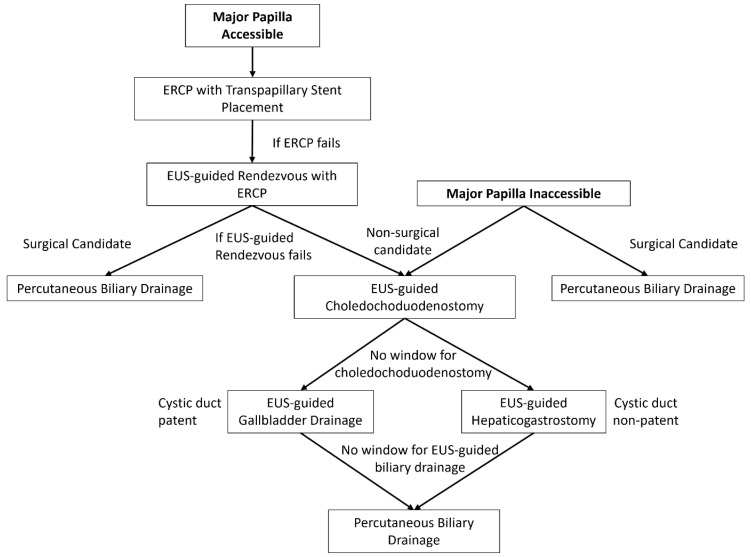
Algorithm for biliary drainage for patients with pancreatic cancer.

**Figure 3 cancers-16-00029-f003:**
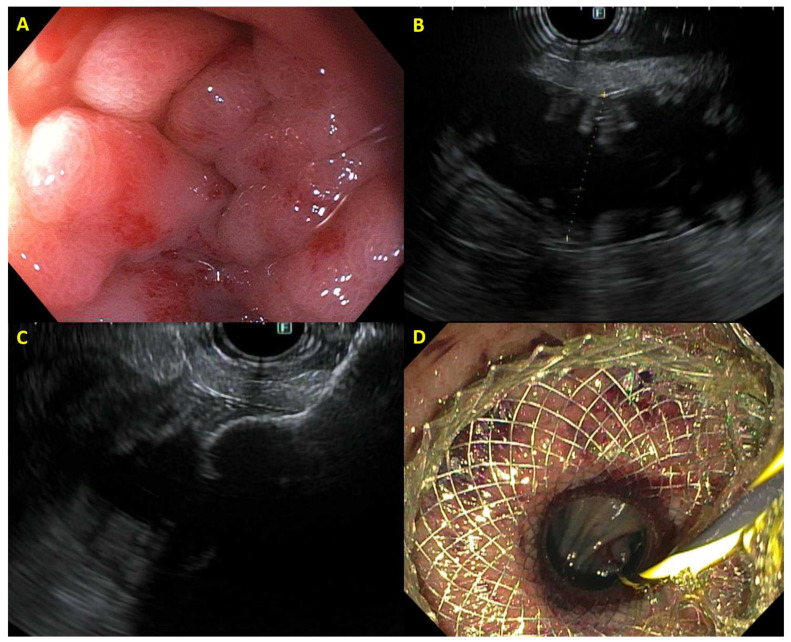
Example of EUS-guided gastroenterostomy with (**A**) malignant duodenal obstruction, (**B**) jejunum identified on EUS, (**C**) distal flange of lumen-apposing metal stent deployed within the jejunum, and (**D**) proximal flange of gastrojejunostomy stent deployed in stomach with jejunum visible through stent.

**Table 1 cancers-16-00029-t001:** Therapeutic Endoscopic Ultrasound (EUS) Indications in Pancreatic Adenocarcinoma.

Disease Complication	Therapeutic EUS Considerations	Adverse Events	Survival Post-Treatment
Biliary obstruction	- EUS-guided Choledochoduodenostomy - EUS-guided Gallbladder Drainage - EUS-guided Hepaticogastrostomy	- Cholangitis (2–7%) - Stent misdeployment (2–5%) - Stent migration (1–2%) - Peritonitis (5%) - Bleeding (1%)	144–232 days (median)
Gastric outlet obstruction	EUS-guided Gastroenterostomy	- Stent misdeployment (0–10%) - Peritonitis (1%)	84–120 days (median)
Afferent Loop Syndrome	EUS-guided Enteroenterostomy	- Stent misdeployment (4–5%)	110–120 days (median)
Pain	EUS-guided celiac plexus neurolysis	- Diarrhea (1–2%) - Fever (1–2%) - Transient abdominal pain (5–11%)	168–314 days (median)

Data obtained from prospective studies described in this manuscript.

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
