# Peer review of "Therapeutic Endoscopic Ultrasound for Complications of Pancreatic Cancer"

_cancers, 2023, doi:10.3390/cancers16010029_

Round 1
Reviewer 1 Report
Comments and Suggestions for Authors
Very interesting and comprehensive review. The authors should comment that there are different kinds of LAMS for these indications, mainly Axios and Spaxus. Data reported in the review are mainly with Axios stent, i suggest to mention some studies also with Spaxus stent (for example cite the recent series PMID: 34339667)
Maybe the authors could propose an algorithm on the use of LAMS in patients with advanced PDAC
When describing RFA, the authors should mention that this therapy is already effectively used for other pancreatic tumors such as insuliinoma (see the recent paper published by Crinò et al in CGH)
Author Response
Response to Reviewer #1 Comments
Very interesting and comprehensive review. The authors should comment that there are different kinds of LAMS for these indications, mainly Axios and Spaxus. Data reported in the review are mainly with Axios stent, i suggest to mention some studies also with Spaxus stent (for example cite the recent series PMID: 34339667)
We thank the Reviewer for noting this. We have accordingly mentioned the other types LAMS examined in the literature.
Maybe the authors could propose an algorithm on the use of LAMS in patients with advanced PDAC
We thank the Reviewer for this suggestion, but found it difficult to comment only on LAMS within an algorithm. We have however, provided a new table that discussed the various therapeutic EUS modalities in the management of advanced PDAC.
When describing RFA, the authors should mention that this therapy is already effectively used for other pancreatic tumors such as insuliinoma (see the recent paper published by Crinò et al in CGH)
We agree with the Reviewer that this is important to mention and have incorporated into the revised manuscript.

Reviewer 2 Report
Comments and Suggestions for Authors
Review
Therapeutic Endoscopic Ultrasound for Advanced Pancreatic Cancer
Samuel Han and Georgios I. Papachristou
Aim
This review will focus on the latest advances in therapeutic EUS for the treatment of PDAC related complications. In presenting data regarding the safety and efficacy of these modalities, we hope to increase awareness of the growing use of EUS as an adjunctive tool for both medical and surgical oncologists in the management of this deadly disease.
General remarks
Well written (me too) review on therapeutic endoscopy and EUS lacking important information for oncologists.
What is new and different to other also well written but almost identical (similar) reports (add on value)?
Strength
Short and precise summary of most (but not all) important studies.
Weakness
Nothing of the reported information is really new and some of the details are somewhat superficially reported without practical impact.
The authors focus on an oncological disease, PDAC, and not on pure “Therapeutic Endoscopic Ultrasound”. Some of the cited references do not focus on PDAC. Please make this clear throughout the manuscript.
The review could add value to oncologists (and also to the journal Cancers), if important evidence based information would be given.
What are the TNM stages in PDAC at time of diagnosis (and life expectancy), especially how often distant metastases can be expected?
How often which complication can be expected to be treated by endoscopy and/or EUS? What happens if not such a treatment would be performed?
At which stage of the disease which complication occur (and the measurable benefit for the patient for each procedure)?
Please consider to create a table to summarize the frequency of complications in advanced PDAC, life expectancy with and without the procedure and additional benefits.
Please add information how medical treatment influences “Therapeutic Endoscopic Ultrasound”?
Major issue
The authors do not differentiate in their report between patients with and without PDAC. The title of the review is misleading and should be changed; example: … (646 days vs. 202 days) ... Very few patients with advanced PDAC survive more than 6 month but 646 days are reported.
Minor issues
Ref numbers are doubled.
Why do you call “ERCP” including “P” but mentioning mostly / only ERC?
Some Refs are cited with large capitals.
The use authors use some expressions, which should be define. Please explain, which objective criteria are used for “landmark” studies and large studies?
Introduction
Traditionally, endoscopic ultrasound (EUS) has played a significant role in the diagnosis and local staging of PDAC [Refs are missing]. All statements should be referenced throughout the paper.
EUS-CD
1. … ERCP may not be feasible in all cases such as when endoscopic access to the major papilla is blocked by extension of the cancer (duodenal obstruction) … How often this happens? Please give numbers (facts and figures) and cite the literature. How long live patients in such a situation (?) (see also below).
2. percutaneous biliary drainage was traditionally offered by Interventional Radiology. Please comment that outside of US often gastroenterologists perform PTCD and other forms of interventions.
3. Please comment on the specifics of the guidewire needed of EUS-CD.
4. … tract is then dilated (with either a balloon, passage dilator, or using cautery) and the SEMS … Please judge the best of the three. There are current papers on controversies in EUS-BDD giving advice.
5. The wording “0% rate” looks odd to me, “no complication” (?)
6. EUS CD had a superior technical success rate (96.2% vs. 76.3%) with no difference in 1 year stent patency rates or other adverse events. Palliative patients with end stage of disease requiring this maneuver die within 6 month after diagnosis or earlier. If true “1 year stent patenty rate” sounds misleading.
7. “The development of the LAMS and specifically the electrocautery enhanced LAMS”. Which systems are established (?); please comment on differences and give references.
8. Figures 1B and 1D show low(er) quality (stent structure (better image recommended) and image quality) on the reviewers screen.
EUS guided hepaticogastrostomy (EUS HG)
In cases where duodenal invasion of the cancer precludes access to the major papilla and the duodenal bulb for performing an EUS CD, EUS guided hepaticogastrostomy (EUS HG) offers another technique for draining the biliary system. As mentioned above how often this happens in the course of the disease (PDAC)? Give references.
· Please comment on the specifics of the best guidewires, dilators and stent technologies needed for EUS-CD.
· … Adverse events specific to the EUS HG include the risk of stent migration into the peritoneum due to the free space between the stomach and the liver and constant “motion of the liver during respiration and mediastinitis”. Please comment why you mention specifically mediastinitis?
EUS Guided Gallbladder Drainage
… biliary drainage modality, a large multicenter study (n=48) from Italy found an 81.3% … Please define “large”, why did you term this study large?
Comparison of EUS Guided Biliary Drainage with Percutaneous Biliary Drainage
If I am allowed to provoke: Misleading chapter, since older studies on PBD reported by real experts are not mentioned. Currently the knowledge and reports of PBD might be biased by loss of expertise.
GOO
I am not quite sure if this reference is really representative: Andtbacka, R.H.; Evans, D.B.; Pisters, P.W. Surgical and endoscopic palliation for pancreatic cancer. Minerva Chir 2004 , 460 59 , 123 136. The chapter seems to me oversized compared to the others (?) and the need for GOO overestimated.
Afferent limb syndrome
I am not sure if this is really a good term. Please explain in more detail (why not loop?).
EUS Guided Radiofrequency Ablation
Please explain from an oncological point of view the possible benefit on survival rate (or whatever)?
Others
Why do you tackle some experimental procedures but not angiotherapy für PDAC complications?
Conclusion
Might be shortened.
Author Response
Response to Reviewer #2 Comments
Aim
This review will focus on the latest advances in therapeutic EUS for the treatment of PDAC related complications. In presenting data regarding the safety and efficacy of these modalities, we hope to increase awareness of the growing use of EUS as an adjunctive tool for both medical and surgical oncologists in the management of this deadly disease.
General remarks
Well written (me too) review on therapeutic endoscopy and EUS lacking important information for oncologists.
What is new and different to other also well written but almost identical (similar) reports (add on value)?
We thank the Reviewer for these comments. We believe the primary add-on value is the focus on a specific disease entity while also including the most recent trials published in this space.
Strength
Short and precise summary of most (but not all) important studies.
We thank the Reviewer for these comments. We have tried to include the most important studies within the focus of this paper.
Weakness
Nothing of the reported information is really new and some of the details are somewhat superficially reported without practical impact.
The authors focus on an oncological disease, PDAC, and not on pure “Therapeutic Endoscopic Ultrasound”. Some of the cited references do not focus on PDAC. Please make this clear throughout the manuscript.
The review could add value to oncologists (and also to the journal Cancers), if important evidence based information would be given.
What are the TNM stages in PDAC at time of diagnosis (and life expectancy), especially how often distant metastases can be expected?
How often which complication can be expected to be treated by endoscopy and/or EUS? What happens if not such a treatment would be performed?
At which stage of the disease which complication occur (and the measurable benefit for the patient for each procedure)?
Please consider to create a table to summarize the frequency of complications in advanced PDAC, life expectancy with and without the procedure and additional benefits.
Please add information how medical treatment influences “Therapeutic Endoscopic Ultrasound”?
- While we agree that none of the therapeutic EUS procedures described are new, we would like to note that there have been several recent multicenter randomized trials published during this calendar year which have solidified the safety and efficacy of these procedures such as EUS-guided biliary drainage and EUS-guided gastroenterostomy with utilization of a lumen-apposing metal stent. We have revised the manuscript to add more practical information for the reader as well.
- We thank the Reviewer for noting that not all subjects included in the referenced studies have PDAC. We have now noted what proportion of subjects from these studies had PDAC.
- We thank the Reviewer for noting this. We have now noted the staging of PDAC in the study populations whenever available.
- The Reviewer makes an excellent point regarding when these therapeutic EUS modalities can be offered. We have now revised the manuscript to make this clearer and have also included a table for the use of therapeutic EUS in PDAC.
- We have now detailed at which stage of cancer progression that the complications (i.e. biliary obstruction, gastric outlet obstruction, etc…) occur and the potential benefit of performing therapeutic EUS in these situations.
- We thank the Reviewer for this suggestion and have now included a table with the requested information, incorporating the suggestions from the previous comment by the Reviewer.
- The Reviewer makes an excellent point regarding how medical treatment (i.e. chemoradiation) affects Therapeutic EUS. There unfortunately is not much data regarding how medical treatment affects the outcomes of therapeutic EUS or the candidacy for therapeutic EUS and the majority of the included studies did not comment on medical treatment details.
Major issue
The authors do not differentiate in their report between patients with and without PDAC. The title of the review is misleading and should be changed; example: … (646 days vs. 202 days) ... Very few patients with advanced PDAC survive more than 6 month but 646 days are reported.
We agree with this point and have removed the word “Advanced” from the title.
Minor issues
Ref numbers are doubled.
Why do you call “ERCP” including “P” but mentioning mostly / only ERC?
Some Refs are cited with large capitals.
The use authors use some expressions, which should be define. Please explain, which objective criteria are used for “landmark” studies and large studies?
We thank the Reviewer for noting these. We agree that ERC is the correct terminology for biliary cases, but given the standard nomenclature of using the term ERCP (including in all the studies referenced), we elected to continue to use ERCP. We have also removed the term “landmark” from our descriptions.
Introduction
Traditionally, endoscopic ultrasound (EUS) has played a significant role in the diagnosis and local staging of PDAC [Refs are missing]. All statements should be referenced throughout the paper.
We appreciate the Reviewer’s point and have provided references for our statements.
EUS-CD
- … ERCP may not be feasible in all cases such as when endoscopic access to the major papilla is blocked by extension of the cancer (duodenal obstruction) … How often this happens? Please give numbers (facts and figures) and cite the literature. How long live patients in such a situation (?) (see also below).
We thank the Reviewer for this insightful comment. We have provided recent data from randomized trials as to how often duodenal obstruction precludes ERCP.
- percutaneous biliary drainage was traditionally offered by Interventional Radiology. Please comment that outside of US often gastroenterologists perform PTCD and other forms of interventions.
We thank the Reviewer for noting this. We have now removed the association of percutaneous drainage with interventional radiology.
- Please comment on the specifics of the guidewire needed of EUS-CD.
We have provided this update in the Choledochoduodenostomy section.
- … tract is then dilated (with either a balloon, passage dilator, or using cautery) and the SEMS … Please judge the best of the three. There are current papers on controversies in EUS-BDD giving advice.
The Reviewer makes an important point, we have now included some comments about methods of tract dilation.
- The wording “0% rate” looks odd to me, “no complication” (?)
We have amended this accordingly.
- EUS CD had a superior technical success rate (96.2% vs. 76.3%) with no difference in 1 year stent patency rates or other adverse events. Palliative patients with end stage of disease requiring this maneuver die within 6 month after diagnosis or earlier. If true “1 year stent patenty rate” sounds misleading.
While we agree that this is misleading, we included data from the primary outcome of the study, which was 1 year stent patency rates. We added the mean survival for each treatment to clarify how long subjects survived post-stent placement.
- “The development of the LAMS and specifically the electrocautery enhanced LAMS”. Which systems are established (?); please comment on differences and give references.
We have provided references for this and have clarified which electrocautery-enhanced LAMS are typically utilized.
- Figures 1B and 1D show low(er) quality (stent structure (better image recommended) and image quality) on the reviewers screen.
We unfortunately do not have a series of better quality images for this procedure.
EUS guided hepaticogastrostomy (EUS HG)
In cases where duodenal invasion of the cancer precludes access to the major papilla and the duodenal bulb for performing an EUS CD, EUS guided hepaticogastrostomy (EUS HG) offers another technique for draining the biliary system. As mentioned above how often this happens in the course of the disease (PDAC)? Give references.
We thank the Reviewer for this comment and have provided data for this within the text as addressed above.
- Please comment on the specifics of the best guidewires, dilators and stent technologies needed for EUS-CD.
- … Adverse events specific to the EUS HG include the risk of stent migration into the peritoneum due to the free space between the stomach and the liver and constant “motion of the liver during respiration and mediastinitis”. Please comment why you mention specifically mediastinitis?
We mentioned mediastinitis as this is a known risk if left hepatic duct puncture is accidentally performed from the esophagus (above the diaphragm).
EUS Guided Gallbladder Drainage
… biliary drainage modality, a large multicenter study (n=48) from Italy found an 81.3% … Please define “large”, why did you term this study large?
We agree with the Reviewer that this is a relative term and have removed the description “large” from this sentence.
Comparison of EUS Guided Biliary Drainage with Percutaneous Biliary Drainage
If I am allowed to provoke: Misleading chapter, since older studies on PBD reported by real experts are not mentioned. Currently the knowledge and reports of PBD might be biased by loss of expertise.
We acknowledge that there certainly may be bias given the changing expertise in PTBD and emphasize that we are only citing data from comparative studies of the two modalities.
GOO
I am not quite sure if this reference is really representative: Andtbacka, R.H.; Evans, D.B.; Pisters, P.W. Surgical and endoscopic palliation for pancreatic cancer. Minerva Chir 2004 , 460 59 , 123 136. The chapter seems to me oversized compared to the others (?) and the need for GOO overestimated.
We thank the Reviewer for noting this and have removed this reference.
Afferent limb syndrome
I am not sure if this is really a good term. Please explain in more detail (why not loop?).
We agree with the Reviewer and have changed this to afferent loop syndrome.
EUS Guided Radiofrequency Ablation
Please explain from an oncological point of view the possible benefit on survival rate (or whatever)?
We have now commented how EUS-RFA may aid in the treatment of PDAC.
Others
Why do you tackle some experimental procedures but not angiotherapy für PDAC complications?
We thank the Reviewer for noting this, but we did not include any angiotherapy as they are not primarily performed via EUS.
Conclusion
Might be shortened.
We have shortened the Conclusion section as suggested.

Round 2
Reviewer 1 Report
Comments and Suggestions for Authors
The revised version of the manuscript is OK. Thank you!
Author Response
We thank the Reviewer for their positive feedback!
Reviewer 2 Report
Comments and Suggestions for Authors
Well improved. Most but not all points tackled.
The development of the LAMS and specifically the electrocautery-enhanced LAMS 84 (Hot AXIOS, Boston Scientific, Marlborough, USA and Hot Spaxus, Taewoong Medical, 85 Gimpo, Korea) has greatly simplified the process of performing an EUS-CD (Figure 1) [15]. Please check if there is / are no other company / companies offering such device. If yes, please add name and company.
Author Response
We thank the Reviewer for noting this and have added the information for a 3rd electrocautery-enhanced lumen-apposing metal stent (Hanarostent Hot Plumber) that is commercially available.